# User-Influenced/Machine-Controlled Playback: The variPlay Music App Format for Interactive Recorded Music

**Justin Paterson [1,\*], Rob Toulson [2] and Russ Hepworth-Sawyer [3]**

1   London College of Music, University of West London, St Mary's Rd, Ealing, London W5 5RF, UK
2   School of Media Art and Design, University of Westminster, Watford Rd, Northwick Park, Middlesex HA1 3TP, UK
3   School of Performance & Media Production, York St John University, Lord Mayors Walk, York YO31 7EX, UK
\*   Correspondence: justin.paterson@uwl.ac.uk

**Abstract:** This paper concerns itself with an autoethnography of the five-year 'variPlay' project. This project drew from three consecutive rounds of research funding to develop an app format that could host both user interactivity to change the sound of recorded music in real-time, and a machine-driven mode that could autonomously remix, playing back a different version of a song upon every listen, or changing part way on user demand. The final funded phase involved commercialization, with the release of three apps using artists from the roster of project partner, Warner Music Group. The concept and operation of the app is discussed, alongside reflection on salient matters such as product development, music production, mastering, and issues encountered through the commercialization itself. The final apps received several thousand downloads around the world, in territories such as France, USA, and Mexico. Opportunities for future development are also presented.

**Keywords:** interactive music; dynamic music; app; remix; automatic remix; commercialization; HCI; music production

## 1. Introduction

This paper discusses the variPlay (2015) system, which was developed over five years, supported by three consecutive rounds of funding from both the UK Digital R&D Fund for the Arts and the UK Arts and Humanities Research Council. This system is an iOS app platform that enables listeners to manipulate musical content as it is played, and to apply intelligent algorithms that deliver a unique version—effectively a remix—upon every listen. These are always representative of the creative vision of the music artist through being populated with artist-approved multitrack audio stems (musical parts e.g., 'drums' with embedded studio processing).

An overview of the pilot proof-of-concept project is presented. This project was the *Red Planet* app by independent artist Daisy and The Dark, with a reflection upon the architecture, design, and operation, including a user-influenced/machine-controlled algorithm. This work entailed some novel approaches to music production and mastering, and these are also considered. The discussion then develops to include issues around the commercialization of the product with project partner, Warner Music Group (WMG), including its impact. Aspects of user behavior, particularly the perhaps gamified response to novel interfaces, also warrant attention.

A number of commercially launched variPlay case studies are also discussed, including variPlay releases by artists: Defab1 (Regent St Records), Ximena Sariñana (Warner Music Mexico),

Ofenbach (Elektra France), and Jack Harlow (Atlantic Records). This is followed by an evaluation of production challenges and opportunities for future development, and then conclusions are formed.

## 2. Evolution of Interactive/Dynamic Music

Recorded music has taken many dynamically variable forms over a considerable time period, from the early days of turntablism through to more contemporary games, app-based systems, and virtual reality. The following section is an extension and adaption of the literature/practice review first produced by Paterson and Toulson (2018c), taking the form of a number of brief chronologies, categorized by theme. Its purpose is to help understanding of how listeners have come to experience interactive-type music and how in turn variPlay drew from this lineage. This will contextualize many of the variPlay features that will be presented in subsequent sections of this text.

### 2.1. DJ Mediation

Many of the musical manipulations typical of variPlay owe their inspiration to long-established approaches such as those developed by DJs. As far back as the 1960s, Osbourne Ruddock (a.k.a. King Tubby) used both faders and mute buttons on a mixing desk to bring elements of a track in and out of the mix, and by patching auxiliary sends and group busses on a mixing desk to dynamically alter the amount of a given signal sent to an effects unit such as a delay or reverb (Partridge 2007). Further, the concept of the 'mashup' describes the playback of two records that are in similar key and tempo at the same time, enabling DJs to blend or multiplex between passages of two songs in a unique and spontaneous fashion (Gunkel 2008).

DJ techniques require very niche skills with vinyl turntables, and of course digital technologies have been developed to simplify and mimic these. Native Instruments (NI) *Traktor* software was launched in 2000 and became recognized as the first digital DJ sync-tool, assisting beat matching and offering new creative opportunities for the digital DJ (Smith 2016). The *Traktor DJ* iPad app was released in 2014 and, in 2015, NI released *Stems*, a DJ tool that offers manipulation of four separate tracks, packaged in an MPEG-4 container (Stems-Music 2015). The *Stems* system works with *Traktor* software, and also includes a content management system that allows musicians to create and package their own material for subsequent manipulation in a live-performance environment. These systems allowed many people fast-track access to the DJ skill set.

### 2.2. Stems

It could be said that regardless of era, DJ music-manipulation was largely conducted by specialists working with stereo audio, but the notion of presenting a wider demographic of consumers with deconstructed audio—stems—is relatively recent. Computer-literate consumers were to become able to repurpose components of commercial music, and public understanding and acceptance of this would follow, a broad concept that variPlay would subsequently build upon.

Rundgren (1993) released *No World Order*, which was the world's first interactive album, hosted on CD-i. This system offered users control over tempo, mood, and playback direction, amongst other things, with content overseen by several eminent producers. Jay Z (2004) officially released an entirely a capella version of *The Black Album*, to encourage remixers to interact with his content and use his vocals with new musical material. Controversially, a (then) unknown remixer, Danger Mouse, chose to 'mash up' *The Black Album* with unauthorized instrumental samples from The Beatles (1968) *The White Album*. The result was called *The Grey Album*, which became a huge (illegal) Internet success (Rambarran 2013). Soon after, in 2005, Trent Reznor of Nine Inch Nails chose to release full multitrack files of the band's commercial releases—legally, so that listeners could process and engage with the music in a novel manner (Reznor 2005).

Since then, bands such as Radiohead (the tracks Nude, and Reckoner, in 2008) have followed suite, although the stems were carefully composited—in terms of their instrumental combinations—to minimize adaptability to illegal destinations. The stems were priced at $0.99 each, and five were

required to fully represent each song (Kreps 2008). Such multitrack audio has since become an established and valuable arts resource for both creative experimentation with music, and for teaching and learning in songwriting/music production (McNally et al. 2019), and numerous 'music-minus-one' apps with stem and tempo control now exist for instrumental teaching and practice, for example, the Trinity Rock and Pop App (Trinity College London 2019). Amongst others, Big Beat Records now release stems and run public remix competitions for most of their artists (Big Beat Records 2019).

A logical extension of these approaches is to provide individual stems to the 'nonspecialist' consumer, prepopulated in an integrated environment that facilitates their mediation. This is the approach taken by variPlay.

*2.3. Democratization of Interactivity*

While the notion of discrete stems changed widespread perception of the makeup of recorded music, it was still only deeply engaged with by a relatively small subset of consumers; those who were equipped with tools and the aspiration to create a purely musical artefact. In parallel, the gamification of interactive music was gaining huge popularity as more emphasis became placed on the interface and a competitive (often social) activity, and the user/player was insulated from perceived 'mistakes'. This ensured that engagement was always fun and did not need to feel like a creative 'task'.

'Generic' video games are likely the medium through which the largest number of people have encountered interactive music. Games have been deploying it since 1981; a game called *Frogger* is generally credited with being the first instance of 'adaptive music' (specifically song switching) by triggering an abrupt musical change in response to a 'game call'—a trigger signal derived from the user's interaction with the game (Sweet 2014). Since then, two principal modes of adaptation have become commonplace: vertical reorchestration, in which instrumental layers are dynamically added or removed according to game state, and horizontal resequencing in which blocks of music are shuffled along the time line, again in response to user activity. Of course, the emphasis is on gameplay, and the music is purely coincidental. In variPlay, both of these concepts were repurposed to make music playback the primary focus.

Music-performance games have proved very popular, with titles such as EA's *Guitar Hero* being launched in 2005 (Activision 2019) and *Rock Band* (Harmonix Music Systems, Inc. 2019) following in 2007. Although the result was music, gaming was the core ethos with control coming from hardware controllers that, in part, emulated real instruments; scores were awarded dependent on how close the player came to the original performance. *Bopler* (TechCrunch 2010) was another interactive-music game—a web-based system that allowed remixing of tracks with a gamified interface and a social media slant, and it featured a number of prominent artists providing content. Peter Gabriel's *MusicTiles* app (Entertainment Robotics 2013), although not an album itself, allowed users to interact with the music and produce unique (albeit basic) mixes of the tracks. Although it apparently aspired to be a music format, it was designed to be more of a game than a playback medium.

Virtual Reality (VR) applications offer several modes of interactivity, from hand-gesture, with/without controllers to gaze, and perhaps represent the greatest momentum in development of current musical interactivity. *Jam Studio VR* (Beamz Interactive Inc. 2017) offers a number of 3-D models of instruments, and the player can trigger phrases that synchronize with a backing track. The concept is that each trigger gesture plays a different phrase, which gives the impression of performance complexity, but the source phrases are long, and the player typically triggers a shorter component section. This 'impression' is also a feature of the variPlay algorithms, where the source material is relatively simple, yet it could be presented in myriad forms. *TheWaveVR* (Wave 2017) emerged in 2017, providing a flexible and sophisticated DJ-esque conceptual-control environment that combined skeuomorphism with VR-specific interface elements. The 2018 *Beat Saber* (Beat Games 2018) requires players to chop flying blocks 'in time' (a much more abstract gesture that *Guitar Hero*) to trigger musical components in an Electronic Dance Music (EDM) style. It has proved to be one of the most popular

VR titles in what is currently proving to be a difficult market for developers—regardless of type of application—and this illustrates the commercial potential for musical interactivity in the present day.

Although variPlay does not attempt to be a gaming platform, the above notion of 'always sounding good', regardless of user interaction is key to its design and operation. As will be seen, the perceived gamification of the app became a significant aspect of its user experience.

### 2.4. Music-Playback Apps

The mobile app format emerged as a most viable platform for releasing interactive music and rich media directly to audiences, and became generally referred to as the 'album app' format (Bogdan 2013). Previously in 2011 Björk had released the first album app, *Biophilia* (Björk 2011), which included a unique interface for music listening as well as custom visual animations, and a modest amount of interactivity with the music. In 2014, Paul McCartney rereleased five of his solo albums as album apps (Dredge 2014). Shakhovskoy and Toulson (2015) further defined a potential album-app platform in collaboration with artist Francois and the Atlas Mountains for their album *Piano Ombre (released in 2014)*, which was recognized as the world's first chart-eligible music app; although these latter two emphasized the popularity of the format, they did not feature interactivity.

Gold (2012) *Tender Metal* album app used a novel algorithmic system to play back a unique synthesized composition on each rendition, without the need for any user control. The *BRONZE* engine that powers *Tender Metal* was developed at Goldsmiths, University of London in 2012 (Gold 2012), and was deployed again in 2016 to play the 24-hour-long *Route One* by Sigur Rós (xl recordings 2016)—without it ever repeating. The *Giant Steps* app (Bauer and Waldner 2013) is one of a family of devices to provide dynamic music for sportspeople; it used the Pure Data language to create music in real time whose tempo relates to the running cadence of a jogger via an iPhone's built-in three-axis accelerometer.

Bernhoft's *Islander* album was released in 2014 as a 2 GB app, incorporating remix and stem-based audio, including interactive features that allowed the listener to manipulate the balance and panning of instrument stems, and to experiment with looping motifs and phrases from the songs (Bernhoft 2014). Its interface was modelled on a simplified mixing desk. The year 2015 saw the *oiid* format launch, which not only allowed volume and panning of stems, but also provided play-along chords and scrolling lyrics (oiid 2015) for a number of artists. This was soon followed in 2016 by Massive Attack's, *Fantom* which remixes songs according to data gathered from device sensors, including the camera, clock, microphone, and the user's heart rate (Monroe 2016). This app aligns with the term 'reactive music', which has been coined by Barnard et al. (2009) to describe music that responds to its environment.

Other stem-player app-based formats include *8Stem* (8Stem 2015) and *Ninja Jamm* (Coldcut 2015), both of which offer extensive sonic manipulation possibilities (typical of music-production workflows with simplified interfaces) albeit requiring significant user engagement, with 'named artists' providing content for the former, and artists from record label Ninja Tune for the latter, which also featured in-app purchases of new music for the interface, a feature added to a later version of variPlay.

Clearly, the app format provided an ideal facility and natural momentum to deploy an interactive/dynamic-music initiative. In fact, the *Piano Ombre* app (above) provided the code base for variPlay's noninteractive features and was pivotal in its subsequent development.

### 2.5. Further Discourse

The term Interactive Recorded Music (IRM) is defined by Paterson and Toulson (2018c) to represent a number of music playback strategies where the listener maintains a level of active control of prerecorded content within a single song. In particular, referring to recorded music that "allows the listener to manipulate musical or technical aspects of the playback [ . . . ] might autonomously vary on each listen [ . . . or] the results of reappropriation after elements of recorded music are made openly available to listener, producer, artist or DJ" (ibid).

The interested reader might consult McAlpine et al. (2009), who produced a review of adaptive music for games; and Redhead (2018), who offered a taxonomy of music descriptors such as interactive, adaptive, autonomous, and reactive. Further, both Zak (2001) and Kania (2006) offered interesting ontologies of rock/popular music, the former with regard to production, and the latter with particular reference to song versions and performance. Consideration of such concepts serves to underpin both meaning and effect of both IRM and other forms of dynamic music.

## 3. variPlay Concept

### 3.1. variPlay Design Overview

The variPlay design allows users to manipulate and interact with commercially released music in ways that have not previously been possible, with the intention of bringing new user experiences and opening a number of artistic and commercial opportunities for music artists and the music industry in general. The concept allows the listener and/or the machine to change the sound or even genre of a given song during playback in a smooth and creative manner. For example, a song with standard pop-rock instrumentation (drums, guitar, synth, strings, and vocals) could be manipulated in real-time to play any of a number of styles, e.g., electronic, a cappella group, acoustic, or a blend of those styles.

In a general context, when a producer renders (also 'bounces' or 'prints') the definitive mix of a song, the constituent audio is likely to be a subset of a greater palette. There might be multiple performance takes, acceptable musical parts that have been editorially muted, different compiled ('comped') edits, parts with different processing or effects, different sequences, or prints of MIDI instruments with different sounds. Furthermore, it is common practice to produce a number of remixes of contemporary pop music releases, and each of these remixes might have a similar palette of bespoke parts, which can be incorporated, if they match in tempo and key. To be utilized in variPlay, all such additional material needs to be rendered into stems that align with a number of designated groupings, e.g., guitars.

Through use of artist-approved stems (rather than digital signal processing (DSP) that mediates the audio and could induce artefacts), the artist and producer have complete control over the musical content in order to attain the aesthetic that they wish. Via the app, these stems might subsequently have their volume levels adjusted, e.g., to rebalance the various instruments, remove the vocals or isolate the drums, thus allowing listeners to create their own preferred mix of the recording. The app can also apply such processes autonomously. The app also contains other rich media that is not typically available with digital-music packages, including artwork, song lyrics, song narratives, producer/performer credits, artist biographies, and direct links to additional media such as online music videos and streaming playlists. This allows the listener to explore the artist's creative world while listening along to their music; an homage to the nostalgia with which 12-inch vinyl sleeves are regarded (Bartmanski and Woodward 2015).

A key design challenge for variPlay was to develop intuitive graphical user interfaces (GUIs) for both the navigation of the album app and also for the interactive manipulation of music, within the design constraints of a mobile-device display. As discussed by Paterson and Toulson (2016), the variPlay navigation is controlled from a home screen that utilizes the main label-approved artwork for the music and displays a menu of selectable features.

The entire variPlay architecture was designed in a modular fashion in order to allow maximum flexibility and customization over different releases. Whilst the various submenus can be configured in development to be bespoke for individual artist preferences, the home menu generally incorporates an item to access interactive music, links to band & artist biographies and production credits, links to image galleries and an 'extras' section holding external links to social-media sites and online media-content. A number of variPlay home pages are shown in Figure 1 to indicate the range of similarities between variPlay apps of different artists, which can carry subtly different buttons.

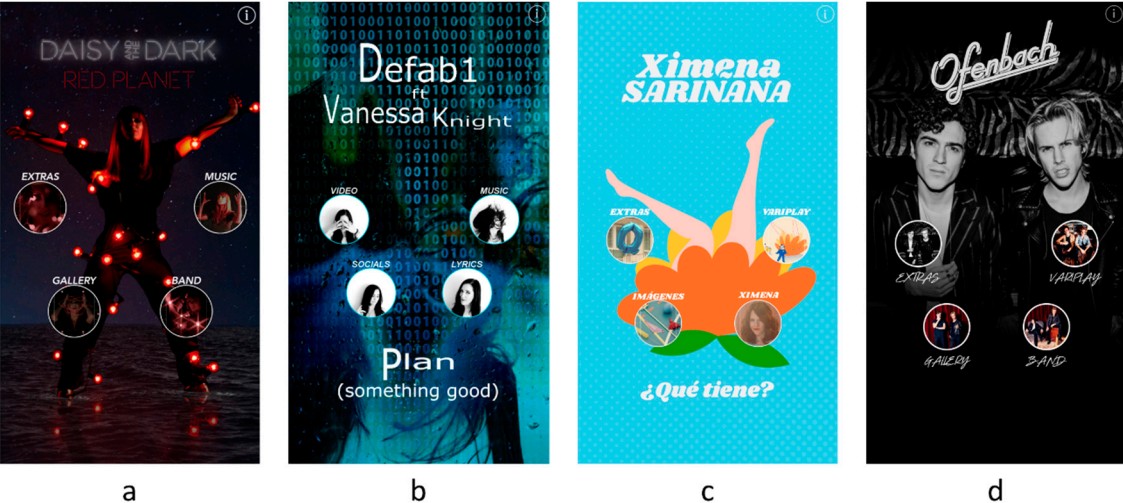

**Figure 1.** variPlay home screens of (**a**) *Red Planet* EP by Daisy and The Dark; (**b**) *Plan* (*something good*) by Defab1 featuring Vanessa Knight; (**c**) *Qué Tiene?* by Ximena Sariñana; and (**d**) *Rock It* by Ofenbach.

Short GIF animations of approximately five seconds with accompanying music from the artist are included in the app, launching when each menu item is accessed. Examples of submenus from a number of variPlay apps are shown in Figure 2.

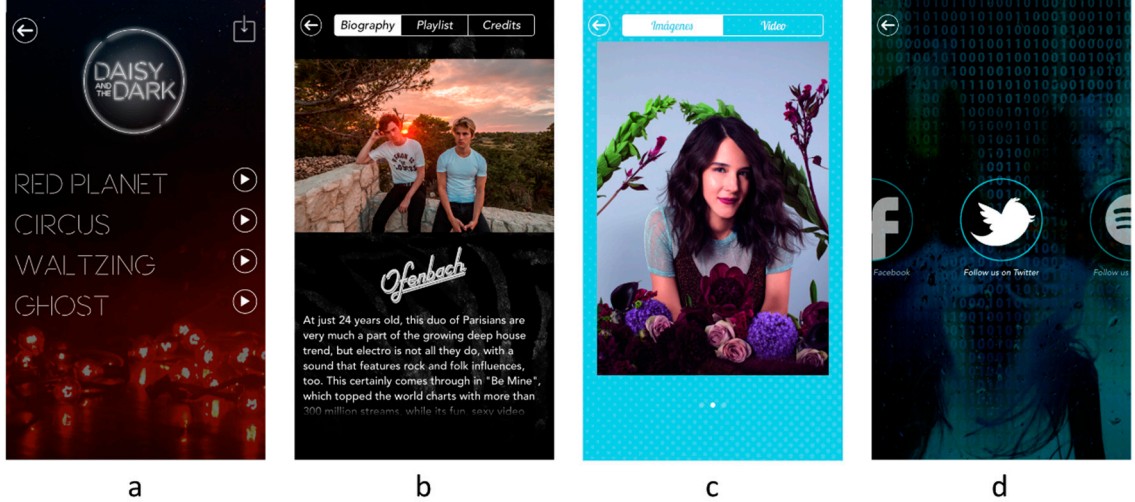

**Figure 2.** Examples of accessible features from released variPlay apps including: (**a**) Daisy and The Dark music player; (**b**) Ofenbach biography; (**c**) Ximena Sariñana gallery images; and (**d**) Defab1 extras section.

### 3.2. Interactive Features

The main IRM interfaces provided by variPlay allow the user to move between alternate mixes in different ways. Two modes of such 'remixing' are implemented, manual and machine-controlled. The manual-remix GUI incorporates a geometric shape (usually a circle), within which the selection of different locations triggers playback of different stem combinations (effectively mixes) of the song; this developed some of the ideas for mouse-based control proposed by Subotnick (2001). Within the geometric GUI is a small circular cursor (the 'mix control'), which controls said selection, and is dynamically repositioned by the user to manipulate playback. For the Daisy and The Dark *Red Planet* interface, the main radio mix of the song is heard when the mix control is positioned in the center of the circular interface, as it is upon launch. Acoustic, electronic, dub and choral interpretations are positioned at 0°, 90°, 180°, and 270° positions on the circle perimeter, as shown in Figure 3. The listener

can choose to move the mix control towards any location or position the mix control at a location which blends between versions in real time. Blending is optimized by background playback of multiple stems simultaneously, each with their own crossfade type (see Section 4.2). Tapping the mix control toggles the vocals on and off, effectively allowing an instrumental version of the song to be played, perhaps for karaoke purposes.

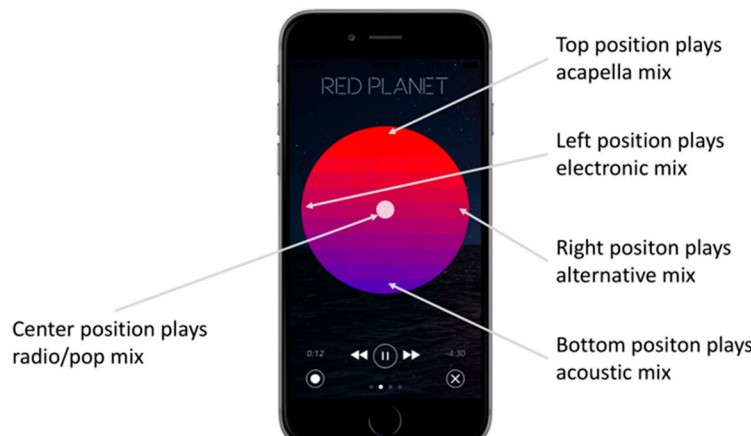

**Figure 3.** Daisy and The Dark 'circle' mixer with annotation (from the variPlay website).

Given that the manual remixer shown in Figure 3 allows virtually unlimited blends between different mixes of the song, which goes beyond the dynamism on successive plays defined by Alferness (2003), it is also possible that some locations of the mix control will sound more musical and viable than others. For example, the blend of two vocal lines in different versions might not actually work well together when blended, giving an unpleasant doubling of voices. This effect can similarly recur for remixes that subtly clash with respect to rhythmic timing (e.g., swing) or dissonance. For this reason, an interface giving access to a curated set of mixes was developed, ensuring that each mix location always utilizes a blend of instruments that have been agreed with the music producer and artist in order to give a high-quality mix at all times.

The matrix interface was also implemented in the pilot project with the song *Circus*, as shown in Figure 4, indicating where a number of specific mixes are situated in specific cells on the grid. When a new grid location is selected by the user, a short configurable fade (see Section 4.2) is implemented algorithmically. The matrix mixer-interface also enables machine-controlled remixing, as will be discussed in Section 4.

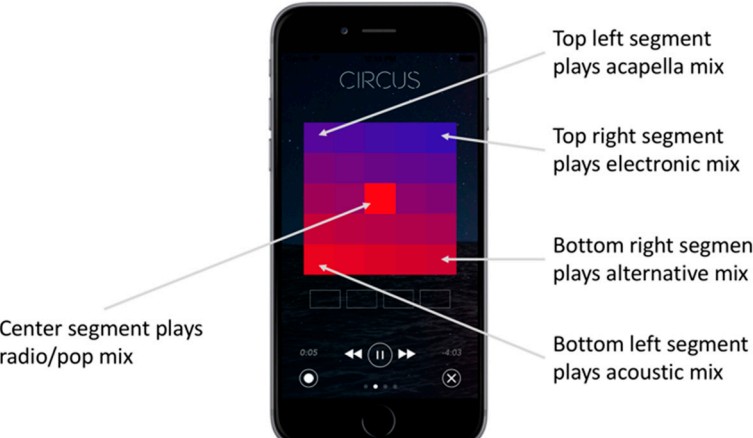

**Figure 4.** Matrix mixer for *Circus* by Daisy and The Dark. Multiple versions and curated mixes of each song are positioned within each segment of a 5 × 5 matrix.

In addition to the remix interfaces, variPlay also enables the user to mute, solo and rebalance individual stems of the playback music, with a simple stem fader interface, as shown in Figure 5, akin to the system introduced by Bernhoft (2014). The user benefit of the stems mixer is to allow an instrumental to be selected, for individual components of the song to be analyzed in detail, or to enable, for example, the drums or vocals to be muted so a user can play or sing along themselves, much as with the subsequently released Trinity Rock & Pop App (Trinity College London 2019).

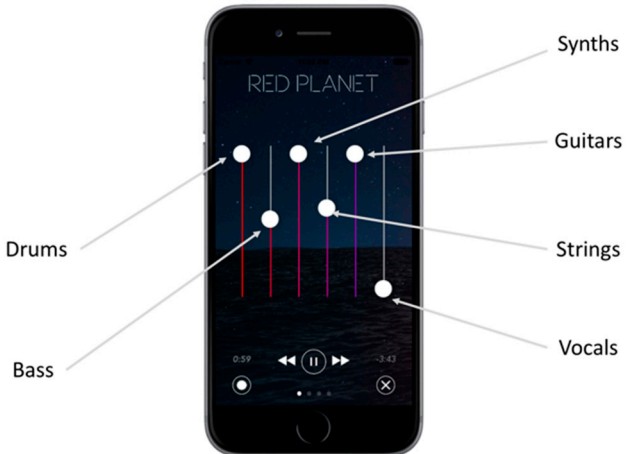

**Figure 5.** Instrument stem faders within the variPlay app. Faders allow mixing, muting, and soloing of musical components.

### 3.3. variPlay Architecture

The functional design for the variPlay playback features is built upon a three-layer architecture. The architecture defines the linking of GUIs with an advanced audio engine, via a rulesets (intelligent audio processing) layer that defines the specific interaction-operations and the user-controlled features, as shown in Figure 6.

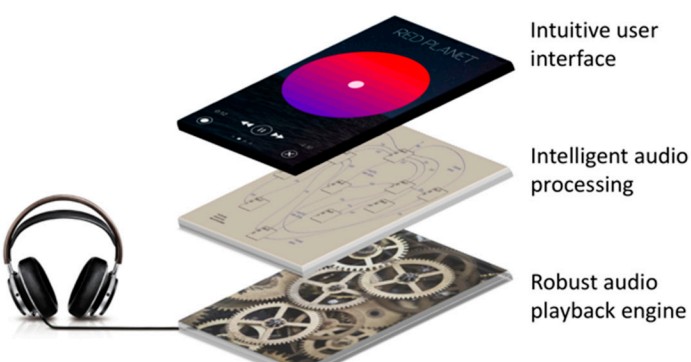

**Figure 6.** variPlay app architecture.

User interaction messages, such as those representing touch locations of screen co-ordinates, button presses, and control-slider positions are evaluated by specific rulesets and used to request corresponding audio-playback and DSP actions from the audio engine. Additionally, the audio engine reports its status to the rulesets layer, which informs the GUI layer to update whenever audio playback is modified, such as a new track starting or when an algorithmic interaction is implemented.

The potential number of stems—especially for an EP release—meant that controlling the package size (hence download time) of the finished app was essential. For this reason, it was decided to implement 256 kbps MP3s. It was found that anecdotally, playing back multitrack MP3s was sonically more gratifying than applying the data compression to a single more complex mix, and the subjective

quality was deemed acceptable for purpose. Stems were printed at 24-bit/44.1 kHz and MP3-encoding was only conducted at the mastering stage (see Section 6.2).

*3.4. Data Mining*

A major benefit of variPlay is that it allows anonymous data mining of listener behavior in ways not captured by other download or streaming music platforms. For example, app analytics enable data about listener preferences with respect to genre and engagement with musical components & structure to be correlated against fan demographics, and can subsequently be used to inform future touring, release, and artist-development strategies. Data analytics were built into the app using the propriety Flurry protocol (Flurry 2019). The data analytics allow collection of user actions for all significant aspects of app engagement, menu selections, song interaction and version preference. It is therefore possible, with enough app users, to gather meaningful quantitative data on the various features (see Section 7.5).

## 4. User-Influenced/Machine-Controlled Playback

*4.1. The variPlay Interface: Automatic Transitions*

The (5 × 5) matrix interface shown in Figure 4 might appear to offer 25 versions of a different song, but these are simply permutations curated from a pool of mix versions, the number of which (in this case) matches the number of rows. One arrangement might be that tapping a cell in the leftmost column plays a given (artist) version with all of its native stems, but each cell to the right substitutes one or more stems with those from other rows, whilst trying to maintain a characteristic sound that keeps a musical coherence across the row (still delivering a multitrack rendition, but with a different musical arrangement or production). Depending on the musical material in a given app, that characteristic sound might be the beat, melodic content or instrumentation. Thus, playing back from consecutive cells on that row gives increasing degrees of musical variation. Accordingly, producing five versions of a song can produce 25 interrelated renditions.

Further, in addition to the related row-cells, through careful arrangement of the stems into 'families' it was possible to define groups of adjacent cells in quadrants of the matrix that could function with musical sympathy. Such an arrangement was employed in the pilot version of the app for the song *Circus*, which had five versions. It placed the radio mix in the center of the matrix, and four alternative versions at the corners. The stem layout for this version can be seen in Figure 7, and tracing the stem-version names gives an idea of typical gradation.

The app can be programmed with a timeline that describes transition triggers which autonomously change the playback cell with precise alignment to the song structure. If these triggers are mapped to cells of a given family with weighted probabilities, then instantiating playback multiplexes between the cells associated with that family, changing the combinations of stems in sympathy with the musical arrangement, with the weighting determining the musical 'center of gravity'. So, although the machine makes playback choices within a given family, the user can select that family according to preference. Thus, the term 'user-influenced/machine-controlled' serves to describe this mode of playback. The effect is that of a song which adheres to a familiar structure and sound, yet with a unique remix upon each listen.

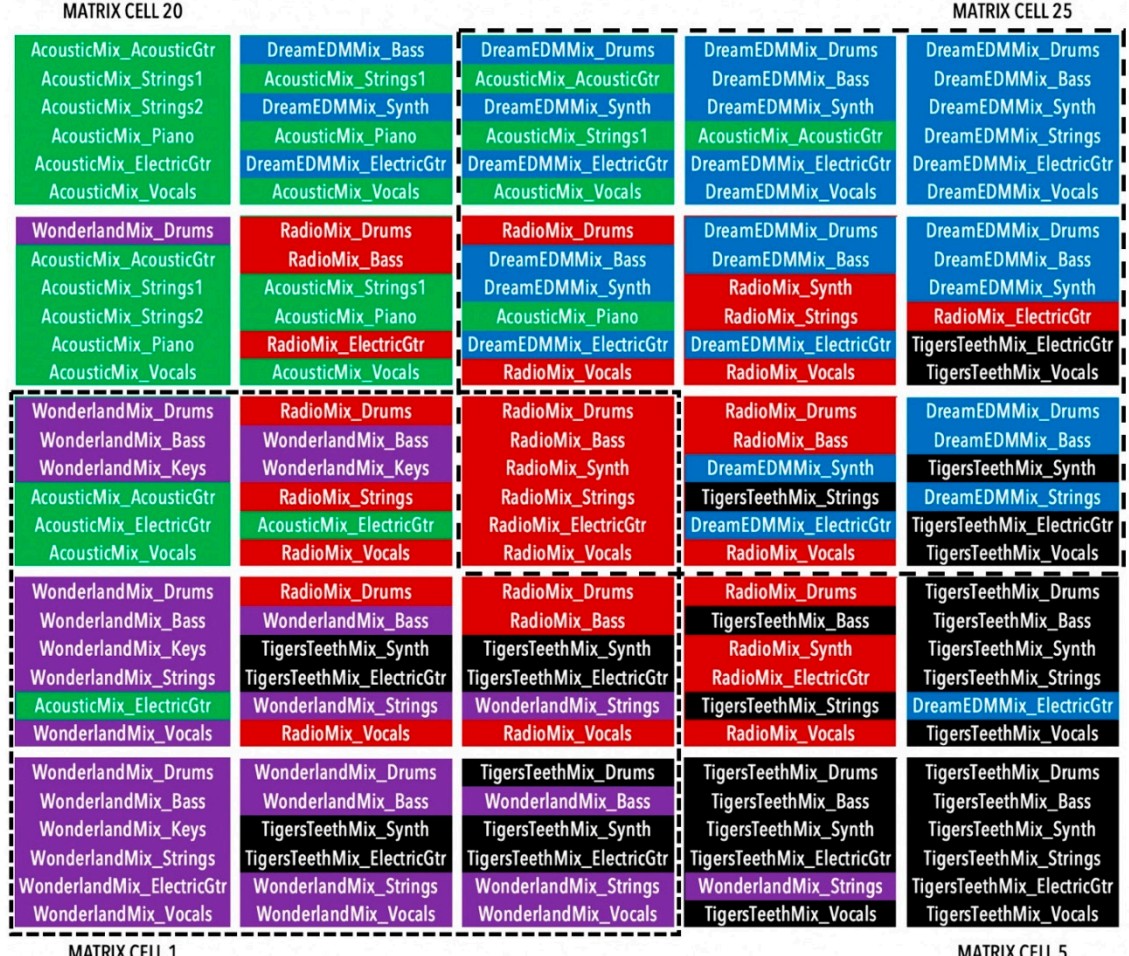

**Figure 7.** The stem layout for *Circus*, with song versions indicated via color coding. Two example families are indicated by hatched boxes, the red-colored radio mix being in both.

## 4.2. Intelligent Cross Fading

In music production, it is commonly understood that different cross fades are required for different types of audio material. For instance, a legato sound such as strings might sound most natural with a relatively slow fade, perhaps with equal-power curves (which help to maintain a consistent perceived volume). On the other hand, a percussion part might benefit from a very rapid transition, preferably occurring between instrumental hits. For each cell transition described above, the audio engine needs to implement cross fades for all of the component stems. The stems of each song version were designated into instrumental clusters, for instance all drum parts resided in a 'drum cluster'. When fading between the mix-version stems of a given cluster, optimal fades were assigned to the cluster's playback channels. Thus, at any given transition point, the multitrack audio appears to change very smoothly as parallel optimal fades are actioned. This applies to both circle and matrix mixers (see Section 3.2).

Despite this, problems can still occur with respect to both pitch and timing. If the source and destination audio files have different pitches—either harmonically or intonationally—then forcing a transition at the trigger point might still sound less than ideal, even with an appropriate fade shape. Accordingly, a system was proposed whereby a polyphonic pitch analysis could offset the point of transition, and instead instantiate the transition at a point of optimal consonance within a gate-time window around the trigger.

Further, staccato material sounds considerably more natural if rapidly switched between musical notes at a point of low amplitude. If a performance has notes that are slightly 'off the grid' (the grid being a series of mathematically precise idealized note time-positions), then attempting to transition

on the grid might inadvertently slice into a drum hit or some such. The dangers of this are exacerbated when transitioning between two 'human' performances, each of which have different microtiming. So, another layer of analysis might determine when a given audio file is below a specified amplitude threshold (or within a specified range), and only when both source and destination meet this criterion, can a transition be actioned. Again, this can be within a specified gate-time window around the trigger point. These concepts formed the basis for a patent application: WO2017068032A1 (Toulson and Paterson 2017).

Another complication comes when different mix versions attribute musically different parts to a given cluster (by necessity or by performance-aesthetic matching). This necessitates a transition that might need to hybridize the various criteria above. Various such permutations are illustrated relative to waveform shape in Figure 8, purely for amplitude in this instance. Given the variety of stems in a given matrix cell, a number of different crossfade profiles are likely to be implemented in a single transition. It was found that a heuristic approach during prototype audio curation yielded the most effective results, although had a more sophisticated hybrid algorithm been developed, machine learning might have offered comparable outcomes.

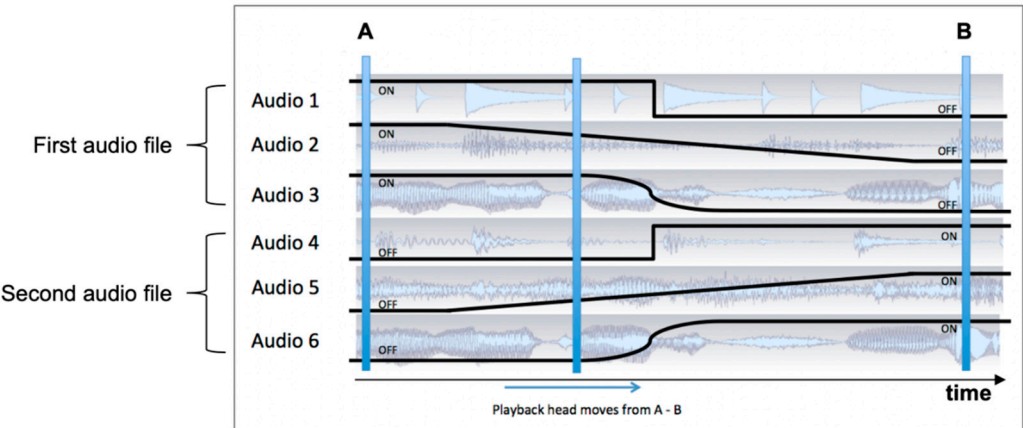

**Figure 8.** For a given audio cluster, transitioning from A to B might involve any of Audio 1–3 fading to any of Audio 4–6, thus demanding pitch and time metadata to optimize timing and shape of the curves.

*4.3. Curation of Audio for Automatic Playback*

4.3.1. Production Concerns

As mentioned in Section 3.1, a core variPlay concept is to only deploy artist/producer-approved audio, rather than dynamic playback achieved via DSP-mediated files that exhibit (even subtle) artefacts when processing. Such audio will likely be drawn from 'offcuts' from the principal production process, but might also be created specifically for variPlay use. There are a number of production and preproduction considerations in each scenario.

The variPlay system potentially presents the most exciting opportunities if the artist and producer anticipate dynamic delivery at the point of creation. The conventional studio production process typically involves considerable trial and error, with many judgements around quality control and aesthetic affecting the trajectory towards the final artefact (Chanan 1997). If instead of only aiming for a single definitive artefact, the producers are mindful of a potential variPlay palette, then audio that might be jettisoned from the final artefact can still put aside to contribute instead to the variPlay system, or even purposefully curated and added to during this process. For instance, a guide acoustic guitar that would not make a final mix could be compressed and effected—and edited if necessary—in a fashion that allowed it to stand alone in a different mix. This would be a relatively time-efficient way of generating new material. Similarly, if two contrasting approaches to a programmed drum track



were working, yet a decision needed to be made in preference to a single one, the less-favored version could once again be put aside for variPlay.

Artists who really wanted to offer the breadth of their expressive range to their fans could deliberately create additional material or even versions of a song in order to demonstrate how it might translate into different renditions, emotions or even genres. This was the approach taken by both Daisy and The Dark and Ximena Sariñana (see Section 5.4).

When preparing multiple versions in a digital audio workstation (DAW)—the industry-standard software used for music recording and production—it proved best to work within a single session with the entire audio palette stacked 'vertically' on audio tracks, subsets of which might form a number of different versions of the song, and it was logical to route the appropriate audio tracks into a single set of group tracks that matched the designated stem groupings. In this configuration, different (overlapping) combinations of tracks can be processed and balanced to form one or more master stereo mixes that can be given to the mastering engineer at a later stage for reference (see Section 6.2). Importantly, meaningful stems can be consistently printed. The word 'consistently' is telling, since in order for the stems to function in different combinations, especially when being machine triggered, they need to maintain a spectral and spatial (the integral panning within the stereo stems) relationship to each other. It was found that sometimes this created a dichotomy in the mixing process, between optimizing a given mix or maximizing the transferability of stems; the latter was generally favored, and the audio fingerprint was always related to the artist mix as a reference.

It is important to note that in either case, mix-buss processing—the simultaneous effecting of the sum of all stems—is best avoided. Whilst many engineers view this as an essential part of the processing chain in order to harness intermodulation-based dynamic and spectral control in particular—to provide 'glue'—the resultant stems will often be played back out of context alongside different musical material. Any such printed buss-processing is embedded with responses to intermodulation from audio tracks not present at the point of 'live rendering'. Such embedding will actually make the stems less tolerant of different playback scenarios. This situation is comparable to object-based audio (OBA), game audio, or multichannel formats that are also rendered at the point of playback. OBA is widely documented although many systems are still in evolution, and the interested reader might consult Silva (2015) who discusses it with regard to television, or Pike et al. (2016), with regard to virtual reality.

When requesting stems from a number of different record labels, it became apparent that the definition of stems was by no means standardized. Although the Recording Academy: Producers and Engineers Wing (2018, p. 13) defines 'stems' as "Any mono, stereo or multichannel surround version created during the mixing process that does not contain all the elements of the Mix Master", in every case, some variation of multitracks, usually just with embedded effects and automation were provided; the variPlay submission guidelines (see Section 5.1) were ignored. This meant that in order to include the artist mix in the apps, a version of that mix had to be recreated, and then the requisite stems derived from it. Further, sometimes only the mastered version of the artist mix was provided as reference. This proved challenging given the issues of above since it was clear that buss processing was commonplace, likely from the mixing stage, but concatenated with the mastering processing.

Although alternative versions were requested, the labels did not usually provide these. Thus, part of the production challenge was to create compatible remixes. This gave the remixers some creative license akin to the creation of bespoke-audio approach mentioned above, and they were able to create some versions to showcase machine-driven playback.

### 4.3.2. Auditioning and Curation

At the outset of variPlay's development, a 'Max for Live' patch had been created to allow rapid prototyping of the app's machine-control algorithms within the Ableton Live environment. Such work lends itself to using 'session view' in Ableton, which allows many combinations to be quickly configured and reconfigured, all mapped into stem clusters (see Section 4.2). This included autoswitching between stem combinations at strategic points relative to the song timeline. The patch

had a number of programmable buttons that could access and test stem families and explore the effect of probabilistic playback weightings in different contexts. It also featured an emulation of the matrix app-GUI to facilitate testing of manual modes on an iPad via Mira (Cycling '74 n.d.).

This patch was subsequently then deployed to test and develop stem combinations in the various commercial app builds. For each new song, once a palette of stems was assembled, it was necessary to combine them in numerous different ways to explore degrees of compatibility and aesthetic possibilities. As had been anticipated, it became clear that groove microtiming was crucial to the range of compatibility of stems. Whereas certain musical parts might have timing 'off the grid' either through human feel or a conscious swing and these might fuse in a given mix to give a good 'feel', when such stems are juxtaposed into new contexts, this did not always work as well, especially when being recombined algorithmically. Consequently, many timing nuances were tightened in individual stems, but naturally, staying as sympathetic to the originals as possible.

It was often necessary to add gain offsets to individual stems in the numerous combinations to increase compatibility and produce useful production effects. It was also found that in unanticipated stem combinations, due to reasons such as frequency masking, relative levels did not always match throughout the duration of a song's arrangement, and so compromise settings were sometimes used in preference.

Each stem in each cell could have a preconfigured transition shape and speed. The transition between cells is calibrated to occur within a 1000 ms window; however, each such stem could be configured to transition at a point, rate and shape which best suits its musical content. Example gain and transition settings for a given cell are shown in Figure 9. Here, it can be seen that the drums and vocal stems are configured to have a quick linear crossfade over a period of 20 ms at the midpoint of the window—in order to allow a quick change from one audio file to another. In contrast, the strings stem is configured to give a much smoother exponential fade over the full 1000 ms transition period.

| MATRIX CELL 12 | GAIN (dB) | TRANSITION TYPE | START (ms) | END (ms) |
|---|---|---|---|---|
| RadioMix_Drums | 0 | LINEAR | 490 | 510 |
| WonderlandMix_Bass | 0 | EXPONENTIAL | 300 | 700 |
| WonderlandMix_Keys | –2 | EXPONENTIAL | 400 | 600 |
| RadioMix_Strings | 0 | EXPONENTIAL | 0 | 1000 |
| AcousticMix_ElectricGtr | +1 | EXPONENTIAL | 400 | 600 |
| RadioMix_Vocals | 0 | LINEAR | 490 | 510 |

**Figure 9.** An example of gain and fade settings for a single cell (cell 12 of the matrix in Figure 7).

## 5. Commercialization of variPlay

Following the initial funding of the research phase, the UK Arts and Humanities Research Council (AHRC) awarded further funding for 'Impact and Engagement' in 2017. This project was partnered by WMG, who undertook to provide financial, legal, organizational, and logistical support towards releasing three major artists from their global roster. Certain aspects of a release might typically be outsourced to a third party, and WMG requested cash support of £10 k/app to mitigate risk for these aspects; this was explained and covered in the grant award.

The concept was to release artists in different genres and/or territories in order to test the market response to the variPlay product. Different pricing models were to be associated with each release.

### 5.1. Submission Guidelines

In order to commercialize variPlay as a format for numerous different artists and genres, it was prudent to produce submission guidelines that standardized the various media formats required to populate each bespoke app. Such media included audio files, images, text, fonts, livery, URLs to artist websites, file-naming conventions, YouTube videos, and preferred Spotify/Apple Music playlists.

Record-label adherence to these guidelines would ensure consistent quality control of the app builds and expedite the design cycle.

## 5.2. Revenue-Generation Strategies

Different pricing strategies were explored with WMG. One approach would be to target an older demographic buying a collectable (app) product of an established favorite band. Such people were likely to have disposable income and be used to parting with money for entertainment products. As such, a variPlay EP app might be priced at $5 or similar. In contrast, a younger demographic engaging with a zeitgeist artist might be more used to free streaming on YouTube or Spotify, and it could be explored whether adding value to the music with dynamic playback might tempt them to part with a small amount of money, for example, $1 per variPlay track. The third option would be to provide a free download, which would clearly attract a greater number of installs, and this could be used as a promotional tool alongside another campaign, for example, an album release or tour.

The variPlay format has screen space (particularly GUI backgrounds) that can be given over to third-party corporate branding. There are options for hosting a persistent logo, or an incremental series of advertising messages across the various pages of the app. Such placement could generate additional revenue or could be used to promote aspects of the commissioning artist without cost, for example tour dates. There is also the opportunity for a 'freemium' model, whereby an initial free download could be followed up with in-app purchases of additional tracks for a nominal sum.

## 5.3. The Royalty-Share Model

One issue that came up in early negotiations with WMG was that of how royalty shares would be split when releasing music as an app created by a third-party developer such as the authors. Whilst there have been many album-app-type releases over a number of years (see Section 2), these have generally been commissioned by the artist's label or the artist themselves. variPlay further complicated this, particularly with regard to publishing, when the music was dynamic and might include remix content generated by others, including the developers themselves. There was discussion around how the licensing might work, and WMG initially preferred licensing the music to the developers at an industry-standard rate, but this seemed counterintuitive to the greater collaborative aims, not to mention expensive.

A potential concern that might be legitimately expressed by artists, was the solo playback facilitated by the fader mixer. This would allow an easy route to unauthorized sampling of components of the track, particularly the vocals. Such practice has led to a number of high-profile court cases, such as Kuos versus Capitol–EMI, which culminated in the transference of copyright of hit pop-song Return to Innocence from the group Enigma [Kuos] (1993) to "an indigenous elderly couple from Taiwan" (Chang 2009, p. 327), whose recording was sampled as the hook of the hit. While this case highlighted that the success of a song could be derived from a significant stem, and therefore potentially requiring attribution of a higher royalty share for such a component, WMG viewed such an approach as overly complex. Further, they shied away from utilizing publicly generated remix content such as that solicited by their label Big Beat Records (2019) for similar reasons.

The solution to guard against sampling was simple; alert artists to this possibility, and either gain endorsement and authorization for any perceived risk, omit the fader feature from the app, or ensure that the stems did not offer a completely solo (e.g.,) vocal—the Radiohead solution (Kreps 2008)—although clearly this limits repurposing of such stems. Another possible issue could be that of moral rights, typically held by the writers, performers, and producers. If the dynamic playback of the app could potentially expose flaws in the performance or recording, this could allow a claim of defamation or parody. Obviously, whilst all stakeholders would be consulted as far as possible in order to gain advance consent, difficulties might be encountered for instance if releasing catalogue material of some age.

It was eventually negotiated that the developers would act as distributors, and their royalty share would be offset against the regular distribution costs. The eventual royalty-share model can be seen in Table 1.

**Table 1.** The royalty-share model.

| Recipient | Comments | Percentage Split of Revenue |
|---|---|---|
| Record Label | Including deductions for artist and aggregator | 35% |
| Retailer | Apple | 30% |
| Music Publishing | Agreement in principle: to be negotiated between WMG and individual artists | 20% |
| Developers | These are the researchers and their parent universities, and other IP holder—Script Ltd. (Shakhovskoy and Toulson 2015). No advance or licensing costs are required. | 15% |
| Total | | 100% |

*5.4. The variPlay Releases: Case Studies*

Following from the pilot release, the commercialization phase involved a number of further apps, each with slightly different implementations of the feature sets which were applied both according to briefs from the various clients, but also through an evolving understanding of user behavior. The first was Plan (something good) by Defab1 ft. Vanessa Knight, Regent St Records (Paterson et al. 2018b). In the playback menu, three interfaces were implemented: a circle mixer, stem faders and a $5 \times 5$ matrix with the 'variPlay' autonomous-remixing mode, and the lyrics were linked to the Home menu, rather than being an item in the mixer pages. The app also included two remixes created by the authors. The next release was Qué Tiene? by Ximena Sariñana, WMG: Warner Music Mexico (Paterson et al. 2018a). Audio-wise, this only featured the circle mixer and stem faders, and it utilized three artist-supplied alternative versions of this song. Independently released EP Chasing Infinity by Asympt Man (2018) only featured stem-fader mixing, but added in-app purchases and playlist linking to the feature set for the first time. Ofenbach's Rock It, WMG: Elektra France (Paterson et al. 2019) included four bespoke remixes, and offered a new mode of 'variPlay playback' that dispensed with the matrix mixer and instead triggered algorithmic playback in one of five styles via a single button for each.

At the time of writing, there is a pending rerelease of '*SUNDOWN*' by Harlow (2018), WMG: Atlantic Recording Corporation. This will instantiate 'variPlay playback' via only a single button to simplify the user experience and emphasize that playback variations are integral rather than excessively controlled by user-GUI choices. On each press of the single 'Hit Me' button, a random family is selected, although the algorithmic rule-sets still operated within it, unseen to the user. It also features four custom remixes and a circle mixer. Figure 10 illustrates the increasing simplification of the matrices over different apps. These are arranged chronologically from left to right.

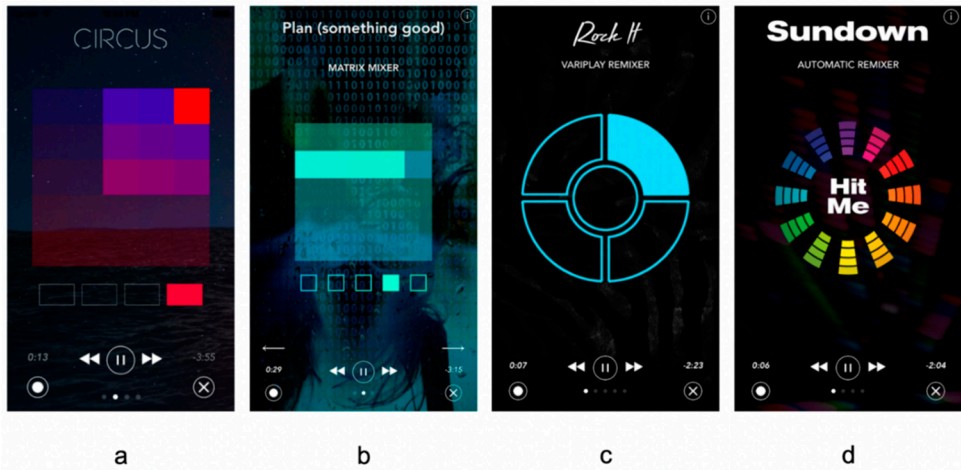

**Figure 10.** (**a**–**d**) Increasing simplification of user control. The colored blocks in the SUNDOWN app on the right are purely visual feedback with no active function.

### 5.5. Commercial Results and Impact

The pilot release was downloaded 2080 times, and as well as the UK, proved popular in China and the USA, and also reached Africa, India, Middle East, and Latin America. Data mining indicated that users spent some 30% longer engaging with the music than with a noninteractive version, released in parallel. variPlay was featured by the Victoria and Albert Museum in London, UK as part of its Digital Design Weekend in both 2016 and 2017, each of which attracted more than 20,000 visitors to the overall events. A bespoke 55″-touchscreen version of the app was developed as an installation for communal public engagement at such events.

The independent releases did not attract significant downloads, although for different reasons (see Section 6.4). However, the British Photographic Institute featured the Defab1 app at its Innovation Hub event (Paterson and Toulson 2018b) and also as 'record of the day' (Paterson and Toulson 2018a).

Of the WMG releases, Ximena Sariñana achieved 1523 downloads (as of 8 July 2019), mostly in Mexico, but also in Bolivia, Columbia, the USA, and Paraguay. The song had 21,048,217 plays on Spotify and 8,718,565 on YouTube (as of 4 July 2019), and peaked on 5 September 2019 at #8 in the global viral-chart. Ofenbach had 984 downloads (as of 8 July 2019), and achieved #1 French Music App, and #103 overall in the French App Store. This song had 7,309,421 plays on Spotify and 4,158,110 on YouTube (as of 4 July 2019). The app was demonstrated on Nick Mason's "The Future of Music Technology: A History of Music and Technology" on the BBC World Service on 22 June 2019 (BBC estimate: 10 m listeners worldwide).

## 6. Challenges with variPlay

### 6.1. Audio Engine Considerations

A given song was specified to use up to six stems, nominally: vocals, guitar, bass, drums, strings and keyboards, although the actual instrumentation is quite possibly different for each song and song version. A maximum of six song versions were also specified. This required 36 song/stem combinations to be available to the user in real time, although only six could be heard at any one point. This was achieved through the design of a 36-track audio player.

The main challenge in designing a suitable audio player was to realize a reliable multistem playback engine. This engine had to enable an exact replication of the original mixes while offering user interaction that seamlessly moved between the curated versions of the songs.

At the time of development in 2014, the available iOS media-player libraries did not accommodate such a complex and precise operation, failing either on their limited playback controls or due to issues to synchronize the required number of stems. For reasons of device consistency, processing

power and multitouch capabilities, the design was solely focused on iOS. Porting the concept to Android-based devices was considered as possible, but time-consuming. Consequently, the bespoke iOS media-player architecture borrowed from the fundamental building blocks available for the design of sequencing applications (including the Audio Unit *AUFilePlayer* library), but time-based scheduling was replaced by robust and sample-accurate synchronization features to ensure that no phase issues would be incurred during playback, fast-forward, and rewind. Each stem was individually controlled in amplitude to facilitate adaptable cross fading, but had its (playhead) synchronization preserved throughout playback across all possible stem combinations.

The specification for the player's control structure was a flexible integration of the GUI design and its associated interfacing approach (see Section 3.3), thus facilitating agile and parallel development of the systems. The aim was to ensure autonomy for the user-interface design by avoiding media-player-specific demands upon it, but to allow for additional rule-based data input that converted user gestures into suitable audio control signals. As these signals potentially required ad hoc and synchronized control over all 36 stems simultaneously, a very accurate and high-priority implementation was developed, governed by stem-specific interpolation rules. For overall convenience during the design and user-interaction stages, all audio and stem-specific data was stored in the JSON file format, which could be used as an overall guide, allowing focus on the sonic and musical relationships between the stems.

### 6.2. Mastering for variPlay

A further challenge for creating variPlay material was mastering. Whilst conventional mastering requires final stereo mixes to be processed for both consistent loudness and sonic attributes between tracks, this approach is not possible for variPlay, given that summation of the instrument stems occurs inside the playback device. This is the channel-versus-object dichotomy (see Section 4.3.1). Melchior et al. (2011) proposed an approach for dealing with this with regard to spatial-audio scenes, and solutions such as Hestermann et al. (2018) are starting to develop. Whilst such future-facing solutions are still emergent, a more pragmatic and immediate solution was required for variPlay.

Consequently, the stems were each mastered individually and designed so that their summation in the device modelled the sound of a stereo master. Figure 11 shows the conventional approach to mastering, with summation of instruments being conducted in the mix studio, and mastering being applied to the stereo mix. Figure 12 shows the approach for mastering five variPlay stems, with mixing resulting in the five stems and mastering being applied to each stem, whilst rendering the summation of the stems.

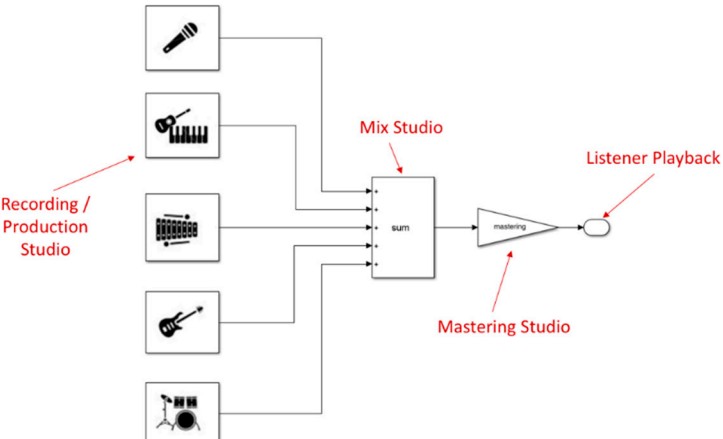

**Figure 11.** Conventional stereo mastering approach.

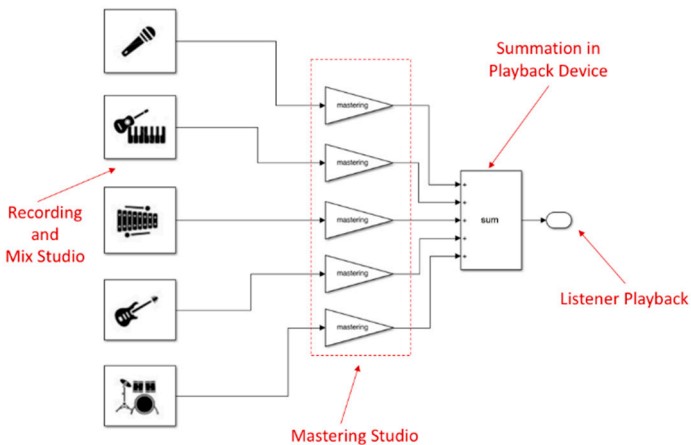

**Figure 12.** Mastering approach for variPlay.

It is useful to note that, in many cases, the record labels provided a conventionally created stereo master of the main radio mix of the song. Where this was the case, that conventional stereo master could be used as a benchmark in terms of dynamics and sonic attributes (spatial and spectral), to inform the variPlay mastering subsequently applied to stems.

However, mastering individual stems also carries considerable disadvantage in that frequency-band build-up/masking and transient summation could be not anticipated in machine-controlled stem combinations, and (akin to the difficulties of mix-buss processing referred to in Section 4.3.1) all spectral and dynamic processing could only exist discretely in stems. Treating the variPlay stems as objects and embedding appropriate metadata would have provided numerous creative mix/mastering opportunities, but this would also have incurred considerable DSP at the point of rendering and so was avoided. This is the approach proposed by Woodcock et al. (2018) specifically for content creators to replicate mix characteristics at the point of rendering. Furthermore, it was not possible to implement passive app-native processing on the output buss due to such CPU-loading considerations, especially for compatibility with older phones.

Whilst a typical modern phone might be able to offer more CPU cycles, the fact that mastering processing is highly specialized means that to gain meaningful and consistent quality, third-party plug-ins would need to be embedded in the app. This would create complex licensing and technical issues. Further, it is commonplace for mastering to utilize hardware tools that have no direct software emulations.

This all meant that a final dynamic limiter could not be used to ensure against unwanted overloads as is conventionally used in stereo mastering. Although limiting was applied to each individual stem, there was a possibility that peaks might align, causing an overload in the final playback when rendered. As a result, more headroom had to be implemented than would be in regular mastering. The CPU issue also precluded integration of a dynamic EQ on the output buss to better control potential frequency buildup, and so the mastering process sometimes erred conservatively in this regard.

Case Study of Mastering Headroom

Table 2 shows the true peak and loudness (LUFS) data for the pilot Daisy and The Dark *Red Planet* EP, which was conventionally mastered as both a stereo EP (by Mandy Parnell of Black Saloon Studios), and as stems for a number of alternate versions of each song for the variPlay platform (by Russ Hepworth-Sawyer of MOTTOsound) using the stereo masters as a benchmark. Measurements of the masters from both engineers were taken to further explore what is predominantly aesthetic creative practice with final levels being determined aurally when 'A/B-ing' between all the various versions. However, it can be seen from Table 2 that, in general, summed-stem versions are approximately 5–7 dB (LUFS) lower than the stereo masters, to avoid risk of peak clipping as above. It is also quite apparent

that different songs require different average loudness to match each other coherently, something that has long been understood intuitively in professional practice.

When working with a number of different artists and genres over subsequent app releases, true peaks were specified to −3 dB to give the mastering process some headroom, with a loudness target of −18 dBFS (0VU); a crest factor—the difference between the two, a subjective indicator of perceived audio quality—of 15 dB. These levels generally appeared to facilitate multiple premastered stem combinations without overload, although the EDM-pop of Ofenbach needed even more conservative levels to prevent overload.

Although the variPlay platform therefore utilizes playback with a lower loudness level than for a conventional music release, it has been the experience of the authors that there were no questions or concerns regarding the loudness of variPlay playback from artists, record label representatives or listeners. Indeed, the greater dynamic range and lack of buss compression required by the variPlay platform appears to give a high-quality sound, free from distortions introduced by compression of intermodulating signals (Toulson et al. 2014), and maintaining clarity of the instruments.

**Table 2.** Comparison of peak and loudness measures for songs and versions incorporated in *Red Planet* EP by Daisy and The Dark.

| Stereo Master (Song Name) | True Peak (dBFS) | LUFS (dBFS) | Summed Mastered Stem (Versions) in variPlay | True Peak (dBFS) | LUFS (dBFS) |
|---|---|---|---|---|---|
| Red Planet | −0.02 | −9.70 | Red Planet (Radio) | −0.34 | −15.13 |
| | | | Red Planet (Acoustic) | −4.29 | −17.61 |
| | | | Red Planet (Dub EDM) | −1.23 | −15.84 |
| | | | Red Planet (Space EDM) | −3.19 | −17.02 |
| | | | Red Planet (A capella) | −3.10 | −18.53 |
| Circus | −0.02 | −9.41 | Circus (Radio) | −0.04 | −15.23 |
| | | | Circus (Acoustic) | −2.15 | −17.19 |
| | | | Circus (Dream EDM) | −1.10 | −15.19 |
| | | | Circus (Tigers Teeth) | −0.41 | −16.41 |
| | | | Circus (Wonderland) | −0.05 | −16.15 |
| Waltzing | −0.08 | −13.34 | Waltzing (Radio) | −2.63 | −20.36 |
| | | | Waltzing (Strings) | −3.08 | −20.02 |
| | | | Waltzing (EDM) | −3.58 | −19.45 |
| Ghost | −0.08 | −11.44 | Ghost (Radio) | −1.10 | −16.69 |
| | | | Ghost (Acoustic) | −1.22 | −16.75 |
| | | | Ghost (Chill EDM) | −4.46 | −18.38 |

*6.3. Gamification*

As an interface design becomes more complex, the user experience becomes less of the intended function, but more of the interface itself (Johnson 1999). This leads to a focus upon the visualization rather than the sonic attributes that the GUI offers. For this reason, the variPlay mixers were designed with a minimal aesthetic. Despite this, during the pilot study, observation of focus-group behavior and interrogation of Flurry data implied that the novelty of the interface intrigued users, and precipitated the question "what does it do"? Users tended to experiment with the controls until they understood that the music was changing, and then appeared to feel that they had mastered 'the game' and then move to another GUI page or disengage. In many cases, users tended to operate the circle mixer very rapidly, too much so to appreciate the actual changes in the music. It was clear that this gestural engagement overrode the desire to actually listen to the music.

Opportunistic data was gathered by watching large numbers of participants at the Victoria and Albert Museum installation. This installation offered only manual controls via virtual buttons on a

large touchscreen, and the music was designed to produce radical remixes that ranged from EDM, through rock to jazz bossa-nova (and more) versions of this song, with instantaneous real-time changes (on the beat) upon touching a button. Some users listened to the music for a while, but most simply wanted to try another button quite quickly. Perhaps most surprising was that about 10% of people tried various buttons, moving through styles often rapidly, and then asked "what does it do"? It appeared to be a matter of perception, of simply not hearing the backing track, but only focusing on the vocals, which played continuously. When they switched to a version with a vocoded vocal, they did notice, and many also seemed aware of dramatic changes of beat, e.g., when a house-style kick drum suddenly came in. A more robust evaluation such as that proposed by Hsu and Sosnick (2009) might have revealed more insight into the HCI issues, but that was deemed beyond the scope of the development at the time.

It was in part for such reasons that successive variPlay releases offered less and less user control on the matrix interface, instead offering more autonomous machine control of musical variations. Whereas the pilot release had four buttons that initiated machine control of a family of stems visible on the matrix, the Ofenbach version did away with the matrix offering five simple buttons albeit with algorithmic complexity behind each. The Jack Harlow variPlay app (the last with WMG) only has a single button that instantiates variable playback, and prompts listener expectation by entitling the page 'automatic remixing'. It does perhaps still tempt over-mediation by labelling the button 'Hit Me', but users need both encouragement and some degree of control if a feeling of interactivity is to be maintained.

*6.4. Political and Logistical Issues*

Any large-scale commercialization of a new technology faces challenges. Although WMG committed to the commercialization phase from the outset, their internal structures are complex. As a global major record company (one of three), they are an umbrella organization for a great many record labels, each one of which is autonomous in its operations, with its own hierarchical team that manages digital affairs and releases. They also have specific liaison personnel assigned to the management companies of individual artists. As a creative tool, it could be said that variPlay is an artist-friendly product, yet alerting artists to its existence involved significant chains of communication, and it proved very easy for conversations to peter out or terminate despite best intentions. Sometimes, even labels cannot communicate with artists since they are entirely buffered by their own management.

6.4.1. Perception of the Product

The user-experiential response to the variPlay platform has been hugely positive, particularly from artists. Sarah Kayte Foster of Daisy and The Dark expressed that the variPlay platform allowed an "experience where the digital listener feels like they have a visual and physical relationship with music again" (AHRC 2015). Responses from artist-liaison personnel of the main WMG releases were also wholly positive from the perspective of delivering new experiences and connectivity opportunities for their artists and fanbases. User feedback from public demonstration sessions with the *Red Planet* EP variPlay app indicated that, of 41 participants, 92.7% of respondents rated the app experience as excellent or good, and 90.3% indicated they would be willing to purchase a variPlay app of a favorite artist.

Although the project aspired to recruit major artists, it transpired through negotiations that their labels and/or management were not often interested in 'experimental/unproven' products, and the conversation often terminated at that point. Rather, some managers might even take the view that such outreach was actually requesting an endorsement of the technology, for which they required equity or up-front advances. This was despite offering them a product that could be sold without any requirement for remuneration. Further, major artists are experienced in large-scale digital campaigns of various sorts via social media or other types of app release, some of which might be live or pending at the time they were approached, and they have come to expect sums of >£100 k to be made available

in order to generate interest. Added to this is the completely understandable scenario where a given artist finalizes a tune and has no wish to see variations of it, and so even when alerted to possibilities, they simply declined as variPlay was superfluous to their aims; this was always understood from the outset of variPlay's development. Further, whilst the original project aspired to release EPs or even albums, it became quickly apparent that the nature of industry forward-planning would preclude this, and instead WMG would only entertain the release of singles.

The industry sets itself targets around streaming numbers and for popular acts, chart position. Although variPlay could act as something to support these, it was not seen as doing so directly, and many industry people simply preferred to invest their time and energy into known routes to primary targets. Given the difficulties that WMG experienced (with their endorsement) numerous other potential stakeholders were approached, ranging from the other major record companies, to large independent labels and numerous artist-management companies. Without direct introductions, few would engage in communication, although both other majors—Universal Music Group and Sony—did, albeit to no avail.

A common response was to attempt to use the technology and its associated funding to support breakthrough artists which by definition do not have established fan bases, and so they were not suitable for the impact agendum determined by the AHRC award. The bespoke nature of variPlay requires largely custom app builds (albeit within a prebuilt technical framework), and record labels sometimes seemed more open to technologies that could be autonomously implemented at scale, especially to invigorate sales of back-catalogues en masse. Another surprisingly common response when describing variPlay's operation, was to be told that they already had tried releasing stems for the public to remix in their own DAWs, often aligned to official competitions, or sometimes were still actively engaged in doing so. It appeared that the word 'stem' had one-dimensional meaning for many.

The Recording Academy: Producers and Engineers Wing (2018) recommend that the format for stem delivery should be negotiated with the mix engineer in advance, but the retrospective nature of most variPlay releases seemed to preclude this, despite providing specific submission guidelines (see Section 5.1). Although this had been anticipated from the outset of the project and funding was provided notionally to pay for correct stem printing (if required), both the chain of communication between labels and production team, tight release schedules, and label desire to focus funding on marketing meant that it did not happen.

### 6.4.2. Promotion

A significant limitation was that in certain territories, e.g., Asia or Latin America, Android dramatically prevails, and by being iOS-only, variPlay could only access a very small market share. Promotion was another issue. Artist buy-in is important, and this must be cascaded to the fans, who have come to expect video via Instagram or Facebook. Whilst the variPlay artists produced promotional videos, one clearly had not used the app before, and another's label insisted that only the currently promoted single sounded over the top of the video footage, making it hard for the viewer to understand what the app might be doing musically. App-Store optimization (ASO) is a practice of making a given app appear to those searching the App Store by generic categories, e.g., remix, and this was not always done as effectively as it might have been, primarily due to cost reasons. Apple did agree to promote the Ofenbach app, but in turn required that the Spotify playlist link was removed, since it clearly competed with Apple Music. A further complication was that due to the evolution of variPlay accruing over three funded research projects that incorporated three universities and various commercial stakeholders, the drafting of a hierarchy of legal paperwork for the commercialization and IP proved protracted and took some 15 months to conclude. Inevitably, and justifiably, this caused the project to lose momentum.

"CPI (cost per install) campaigns are specific to mobile applications. In a cost-per-install campaign, publishers place digital ads across a range of media in an effort to drive installation

of the advertised application. The brand is charged a fixed or bid rate only when the application is installed." (What Is CPI? 2019)

CPI is calculated by total spend on advertising divided by the number of installs, and in 2018 (last available figures), average CPI statistics indicated:

- iOS app CPI globally—$0.86
- iOS app CPI in US—$2.07
- Cost Per Install on Facebook Ads—$1.8
- Cost Per Install on Twitter Ads—$2.53
- Cost Per Install on Instagram Ads—$2.23

(CPI Rates 2018)

Whilst it is not possible to divulge specific commercial budgets for this project, it is clear from download numbers versus outlay (see Sections 5 and 5.5) that the variPlay CPI is significantly beyond more generally accepted standards.

Another unexpected challenge was around the release of the DeFab1 app. The app was almost complete, with a marketing campaign in place aligned to a release date that was tied to the regular stereo version. A few days before that date, a new version of iOS came out, meaning that the code had to be updated for compatibility, and the app then had to be resubmitted to Apple for approval prior to placement on the App Store. This approval was only forthcoming after the release date, which had a significant negative impact on the marketing campaign and subsequent uptake.

## 7. Future Possibilities

### 7.1. Phrase Juxtaposition and Looping

In prototyping, a dynamic version of the audio engine was developed. Whereas the engine that was deployed commercially utilized a playhead that was locked to the same sample-accurate time base for all stems simultaneously, the dynamic version allowed multiple playheads on all the stems. By detecting structurally important parts of a given stem, random-access playback allowed juxtaposition of phrases such as drum fills and instrumental renditions of entire verses, etc. These could be deterministically instantiated to give the impression of a unique performance, akin to that of a live band. If desired, all stems could behave like this simultaneously with completely different trigger points. This led to some interesting creative musical possibilities, akin to an algorithmically controlled sampler. Efforts were made to develop a simple user-content-creation system, and were based around defining phrases and song sections (per stem) with MIDI notes in a DAW—outputting a MIDI file for subsequent decoding—since a ubiquitous system was required for maximum compatibility with diverse user systems, and use of such a system would greatly aid prototype development. An example view can be seen in Figure 13. This proved too inaccurate to deploy at fully sample-accurate resolution, and development was sidelined, although it showed great potential for a future project.

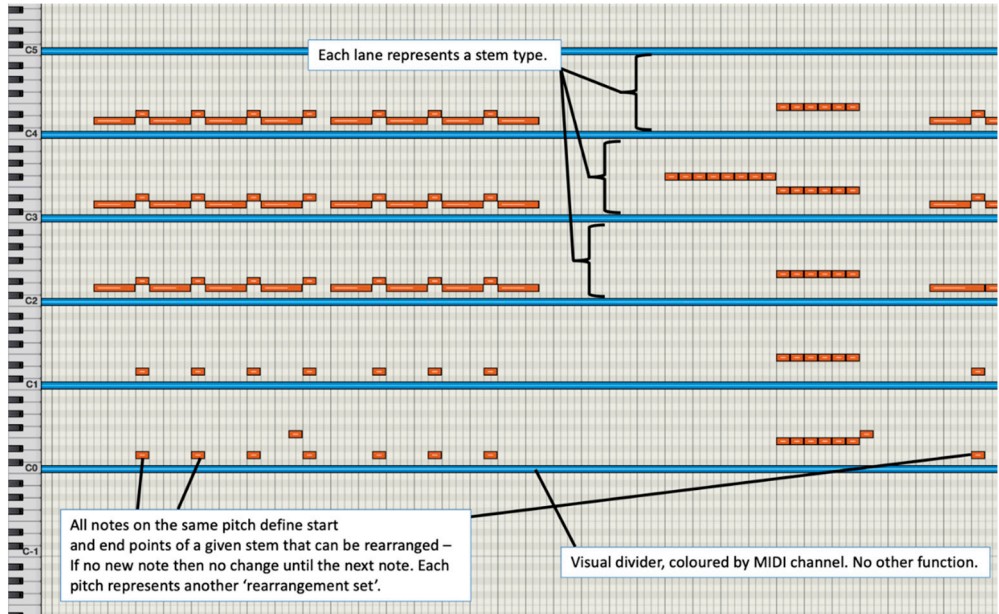

**Figure 13.** The digital audio workstation (DAW)-based piano-roll view of how a user might define time windows for juxtaposition.

A variant on this system was prototyped that was based upon principles that subsequently came to be associated with OBA via the Orpheus project (Silzle et al. 2017), whereby the song could be split into metrically matched sections, and made to play for different lengths of time using looping and skipping. This could allow easy deployment of a stinger-length version, or a 12-inch-type extended play.

## 7.2. Visual Enhancements

With the exception of the GIF screen transitions and buttons, and the 'comet tail' of later circle mixers, the GUIs on all of the apps are rather static, and it is acknowledged that a more engaging user experience could be precipitated by incorporating a more animated interface. This would need to be done with care due to the reasons outlined in Section 6.3.

## 7.3. Content Management System

Given the time associated with building each new app individually, it was clear from the outset that a content management system (CMS) by which users might populate their own apps would expedite future development. During the first variPlay project, a CMS was prototyped in Max, based around the Dict object. This allowed users to easily assign stems to cells and families and set any gain offsets and transitions (see Section 4.1). It was anticipated that the ultimate deployment would be browser based. Implementation of Blockchain technology at this point could offer users a 'genesis' block (Zheng et al. 2018) to further protect stems prior to release.

However, during commercialization in the final phase, further research was not funded, and it was instead decided to focus upon manual construction of apps for the small number that needed to be built. In the future, a CMS would be a most useful tool.

## 7.4. Mastering Levels

Examination of the levels when mastering implies some interesting findings. It is well understood that professional judgement guides mastering engineers' balance decisions, and these rightly supersede numerical readings for loudness, non-clipping peak level and crest factor. A further study might investigate the relationship between the aesthetic of the mastering levels over different tunes (and their instrumentation, arrangement, mix and sequence) and the numerical results around loudness and

crest factor. Drawing comparisons between conventional and stem mastering would provide further insight. Such a study might draw from beyond variPlay, and indeed have a much wider influence.

*7.5. Data Analytics*

The analytics were both intended to inform record label understanding of consumer preferences, and to inform future development of the app's GUI and functionality. Due to the relatively small download numbers, the labels did not seem interested in the data, and so it was not collated and analyzed. However, there still exists the opportunity to perform detailed qualitative analysis, and this could form the basis for a further strand of study.

Another feature that was prototyped was the sharing of individual user mixes on social media or direct email via a small data file. Users would be able to export and import these and hear each other's favorite versions. There was no real appetite for this from the labels, and due to the associated complexities, it was not implemented, but could make a useful feature for the future.

## 8. Conclusions

This five-year project commenced with the development of an innovative delivery format that offered dynamic playback, with both machine and user control. At the time of inception in 2014, apps—as a meme—were very popular, and a number of comparable apps came into being, perhaps even influenced by variPlay. Throughout the subsequent phase of variPlay's commercialization, 'boutique' apps have become less popular with the public, and the music industry now regards them as slightly burdensome, formally associating (marketing) CPI to 'drive' installs, with the variPlay CPI proving prohibitive. Further, charging for downloads appears to greatly lessen uptake. Subsequently, the business model is very much of the app being something value-added to conventional delivery of music to the consumer, but development and marketing costs mean that apps often effectively run at a loss.

Although download numbers were not as high as hoped, WMG seemed very pleased with them. This is a testament to their market understanding and industry appreciation of the concept and product. Both they and the other two majors continue to release various forms of music apps, often with very large budgets for major artists who purposely commission them. Nonetheless, it does underline the need for significant investment, either internal or external, in order to drive these, and that expectation is low.

Whilst it is still entirely possible that if a sufficiently impactful artist were to release music in the variPlay format, it would become a most desirable commodity, perhaps subsequently attracting a cash value, yet the greater music industry—particularly in the UK—has a reticence around this, and seems unwilling to take what it perceives as a high-profile risk with such artists. Overseas territories seem less conservative, although it is still difficult to gain traction, not least whilst communicating through complex industry structures. Android support is especially necessary in many overseas territories.

Despite this, there still seems to be an appetite for commercialization of aspects of interactive or dynamic music within the industry, but this would need to be fully automated in order to operate at scale in order to deliver significant financial return. As such, there are opportunities for artificial intelligence to define future approaches, perhaps through feature extraction, blind source separation, and audio repurposing.

It might appear that consumers have difficulties with disruption of the passive practice of listening to music. Presenting them with an interface immediately triggers a performative expectation, a distraction that subverts the actual music. Machine control can mitigate against this, and branding such control as 'remixing' might yet gain further acceptability.

The technology itself might be viewed as a successful outcome in that it functions as specified, attracted commercial support, has been downloaded by several thousand people, obtained a significant media profile, and a series of releases have evolved variPlay into a brand. Much has been learnt about the process of working with 'music and the machine' at a commercial level. It is hoped that this article

will offer insight to those who might pursue similar trajectories, allowing them to anticipate difficulties and optimize opportunities, to place interactive recorded music into daily life in the future.

**Author Contributions:** Conceptualization, R.T., J.P.; Methodology, R.T., J.P.; Software, R.T., J.P.; Data Curation, R.T., J.P.; Music Production: R.T., J.P.; Mastering: R.H.-S.; Writing—Original Draft Preparation, R.T., J.P.; Writing—Review & Editing, R.T., J.P., R.H.-S.; Project Administration, R.T., J.P.; Funding Acquisition, R.T., J.P.

**Funding:** This research was funded in three tranches by: (1) "Script": funded by Digital R&D Fund for the Arts: Nesta, Arts and Humanities Research Council and public funding by the National Lottery through Arts Council England: https://webarchive.nationalarchives.gov.uk/nobanner/20161103232749oe_///artsdigitalrnd.org.uk/projects/script/; (2) "Transforming Digital Music: Investigating Interactive Playback": funded by AHRC. Dates: Oct. 2014–July 2015. Project Reference: AH/M002535/1 Gateway to Research: http://gtr.rcuk.ac.uk/projects?ref=AH%2FM002535%2F1; (3) "The Commercialisation of Interactive Music": funded by AHRC. Dates: Jan 2018–May 2019. Project Reference: AH/R004757/1 Gateway to Research: http://gtr.rcuk.ac.uk/projects?ref=AH%2FR004757%2F1.

**Acknowledgments:** Warm thanks go to the numerous collaborators in this extended project, principally including: The commercial artists: Daisy and The Dark, Asympt Man, Defab1, Ximena Sariñana, Ofenbach, Jack Harlow; Additional musicians: Sarah Kayte Foster, Brian Miller, Matt Doherty; Additional software developers: Sebastian Lexer, Tim Webster, Steve Massey (Freezabox); Additional remixers: Martyn Phillips, Mike Exarchos (Stereo Mike), Steve Massey (Freezabox); Additional mastering engineer: Mandy Parnell (Black Saloon Studios). Commercial partner: Warner Music Group and their Head of Creative and Digital, Josh Saunders; Endorsed by the British Phonographic Institute; Copyright: All figures were created by the authors.

**Conflicts of Interest:** The authors declare no conflict of interest.

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
