# Peer review of "User-Influenced/Machine-Controlled Playback: The variPlay Music App Format for Interactive Recorded Music"

_arts, 2019_

Round 1

Reviewer 1 Report

The app described in this paper represents an interesting case study in terms of music technology. Unlike many technologies described in academic papers, the authors have worked with musicians and the music industry to develop a potentially marketable project.

While the authors had provided a lot of useful insight from the label and artists’ perspective, it would be useful to have more specific information on how users actually engage with the app. Is this a more “interactive” way of listening to the artists’ material through engagement? Does the user have the capability to create a remix that is creatively unique (e.g. the “chopped and screwed” style)? Can users share their remixes? More attention to how users engaged with the app, perhaps including a qualitative study, would be provide a much deeper level of meaning as to why an app like this could become commercially successful.

-Abstract: “Interactive” music certainly predates turntablism—while I can understand this as a departure point for the article, the way this fact is stated in the abstract sounds inaccurate. Can the authors provide more context as to why they begin with turntablism?

-Section 1 needs to be copy edited.

-The “precursors” section (2.1) seems to conflate “open” music (chance music, aleatoric music, etc.), where the musical results from a given set of instructions are different for each performance, and “interactive” music, which is not well defined in general, but seems to refer to the interaction with electronics to produce sound. There are many, many secondary sources on both of these topics that might help clarify the specific aspect of the author’s discussion. More simply, I’d suggest focusing on the specific musical precursors to the app under development, without getting into the whole history of open music or interactive music.

-It would be nice if the background section that relates to the authors’ app had some context—perhaps it could be mentioned that the specific items discussed in section 2 relate to specific functionality of the authors’ app?

-The definition of “interactive music” should occur before several different categories of interactive music are discussed in section 2. This seems like an error in section 2.6 which refers to “IRM” before this term is introduced in section 2.7.

-In general the purpose of section 2.6 is unclear. The point about adaptive music and video games relates well to the authors’ app (and to apps in general), but why is there so much detail about VR? How is “Beat Saber” that different from its predecessors “Guitar Hero” or “Rock Band”?

-There’s a whole history of generative music (which perhaps is more appropriate background than “open” music) that the authors could discuss that would lead to adaptive music mentioned in 2.7

-There are many missing references beginning at the bottom of p.5

Author Response

I thank the reviewer for some very useful observations.

“There are many missing references beginning at the bottom of p.5”

I queried the missing references and received the following response, which requires no further action from me:

Thank you for your message. The "missing references" refers to the
citations of the figures in the text. This problem does not influence
the article content, and it might caused by the procedure of PDF
conversion. We will inform the reviewer about this problem to make sure
we can solve it after your resubmission.

Please do not worry about it, and if you have any question, please let
me know.

Kind regards,

Ms. Chloe Li
Assistant Editor
E-Mail: 
[email protected]

“Is this a more “interactive” way of listening to the artists’ material through engagement?”

I hope that Section 3.2 is now a little more explicit regarding user operation.

“Does the user have the capability to create a remix that is creatively unique (e.g. the “chopped and screwed” style)?”

No. See section 3.1: “Through use of artist-approved stems (rather than Digital Signal Processing (DSP)—that mediates the audio and could induce artefacts…)” Also, there is now explicit reference to volume levels (only).

“Can users share their remixes?”

No. See section 7.5.

“More attention to how users engaged with the app, perhaps including a qualitative study, would be provide a much deeper level of meaning as to why an app like this could become commercially successful.”

This was an aspiration of the variPlay project, but was not done beyond the pilot project’s focus groups etc. As is reported in Section 7.5, it was an aspiration to analyse the meta data, but this was not done in the end.

“Abstract: “Interactive” music certainly predates turntablism—while I can understand this as a departure point for the article, the way this fact is stated in the abstract sounds inaccurate. Can the authors provide more context as to why they begin with turntablism?”

“Turntablism” removed from abstract and migrated to Section 2 where it is discussed more succinctly.

“Section 1 needs to be copy edited.”

Some Section 1 proofing done, but the publisher will ubiquitously copy-edit further.

Regarding Section 2:

Two of the reviewers requested adjustment of Section 2, but in different and sometimes opposing ways – I was asked to both contract and extend this section. In an effort to find middle ground, it was not possible to fully implement all suggestions. However, several blocks of text have been deleted, and the briefest amount of contextualization has been added with reference to variPlay in an attempt to give a more narrative quality to underpin relevance whilst still remaining fairly succinct, as requested. I hope that this is deemed satisfactory overall.

Reviewer 2 Report

In terms of royalty-sharing, I would suggest implementing Blockchain technology (using smart contracts) as a trustless way to embed the ownership rights of each stem file.

In terms of royalty split, the vocal and or lead instrument stems that carry a clear defined melody and or riff should receive a larger share of the split compared to other stems, especially if for example a resultant song was built around the vocal stem. A good example of why this is needed is the controversial legal case against the German pop group Enigma over the song Return to Innocence (1994), which should have been cited in this paper. Kindly refer to https://www.jstor.org/stable/41225259?seq=1#page_scan_tab_contents

Author Response

I thank the reviewer for some very useful observations.

“In terms of royalty-sharing, I would suggest implementing Blockchain technology (using smart contracts) as a trustless way to embed the ownership rights of each stem file.”

I have added a reference to how blockchain might be implemented in the future in Section 7.3.

“In terms of royalty split, the vocal and or lead instrument stems that carry a clear defined melody and or riff should receive a larger share of the split compared to other stems, especially if for example a resultant song was built around the vocal stem. A good example of why this is needed is the controversial legal case against the German pop group Enigma over the song Return to Innocence (1994), which should have been cited in this paper. Kindly refer to https://www.jstor.org/stable/41225259?seq=1#page_scan_tab_contents”

I have added this useful Enigma reference: see paragraph in Section 5.3. However, the actual royalty split that we negotiated with Warner Music Group is post facto. They certainly shied away from complexity, and nothing can be changed at this stage. It might be a useful point to raise in the future.

Reviewer 3 Report

I think some of the discussions of technical aspects may be somewhat specialized for the general readership of this particular journal, especially those surrounding acoustics and production. A few clarifying sentences here and there may help. Some terms need explaining: DAW, "off the shelf", etc. I wonder whether "remix" is the right word in certain instances when stems are being created with the intention of being combined in different ways? It was not until page 5 when I encountered "artist-produced stems" that I really understood the nature of the app. I think this needs to come sooner. The discussion of stems and remixing raises additional and interesting questions about the ontology of the "song" that could have been further explored (although there were nods to this). There is a considerable body of musicological scholarship on the ontology of musical "works" and "versioning" that seems relevant. Section 2 seems a bit rushed and, simultaneously, a bit disjointed and not always immediately relevant to a discussion of the app. I think a discussion of remix history could be expanded--see for example Chapter 1 of Mark Butler's Unlocking the Groove. But overall I think a more streamlined and effective way of organizing section 2 could be found. I found this article to be very informative in addition to providing a detailed overview of the app and many of the issues confronted in its development. It touched on many interesting matters regarding music technology, the music industry, and music creation. I could see myself recommending this article or discussing the app in courses or contexts ranging from practical music technology/composition settings to media studies courses.

Author Response

I thank the reviewer for some very useful observations.

“I think some of the discussions of technical aspects may be somewhat specialized for the general readership of this particular journal, especially those surrounding acoustics and production. A few clarifying sentences here and there may help.”

Various technical terms such as “mix-buss processing” and “equal-power curves” have now defined in situ.

Some terms need explaining: DAW, "off the shelf", etc.

DAW has now been defined in Section 3.1 and “off the shelf” rephrased in Section 4.3.1. Various other terms have also been expanded.

“I wonder whether "remix" is the right word in certain instances when stems are being created with the intention of being combined in different ways? It was not until page 5 when I encountered "artist-produced stems" that I really understood the nature of the app. I think this needs to come sooner.”

“Artist-approved multi-track audio stems” and “effectively a remix” are now mentioned in the introduction in order to prime the reader from an early stage. Remixing is related to ‘versions’ in the introduction.

Section 3.2 now has scare quotes on remixing to introduce that process in the context of the app.

“The discussion of stems and remixing raises additional and interesting questions about the ontology of the "song" that could have been further explored (although there were nods to this). There is a considerable body of musicological scholarship on the ontology of musical "works" and "versioning" that seems relevant.”

Section 2.5 now has a specific reference to ontology of versions, however, I feel that attempting to address the ontology of songs/versions would quickly require copious amounts of text and discursion has been kept succinct: see below.

Regarding Section 2:

Two of the reviewers requested adjustment of Section 2, but in different and sometimes opposing ways – I was asked to both contract and extend this section. In an effort to find middle ground, it was not possible to fully implement all suggestions. However, several blocks of text have been deleted, and the briefest amount of contextualization has been added with reference to variPlay in an attempt to give a more narrative quality to underpin relevance whilst still remaining fairly succinct, as requested. I hope that this is deemed satisfactory overall.